# Inside Perspective of the Synthetic and Computational Toolbox of JAK Inhibitors: Recent Updates

**DOI:** 10.3390/molecules25153321

**Published:** 2020-07-22

**Authors:** Adriana Coricello, Francesco Mesiti, Antonio Lupia, Annalisa Maruca, Stefano Alcaro

**Affiliations:** 1Dipartimento di Scienze della Salute, Università “Magna Græcia” di Catanzaro, Viale Europa, 88100 Catanzaro, Italy; a.coricello@unicz.it (A.C.); francesco.mesiti@studenti.unicz.it (F.M.); maruca@unicz.it (A.M.); 2Net4Science srl, Campus Universitario ‘S. Venuta’, Università ‘Magna Græcia’ di Catanzaro, Viale Europa, 88100 Catanzaro, Italy; antonio.lupia@net4science.com

**Keywords:** Janus kinase (JAK), multitarget, inflammation, cancer, synthetic strategies, tofacitinib, ruxolitinib

## Abstract

The mechanisms of inflammation and cancer are intertwined by complex networks of signaling pathways. Dysregulations in the Janus kinase/signal transducer and activator of transcription (JAK/STAT) pathway underlie several pathogenic conditions related to chronic inflammatory states, autoimmune diseases and cancer. Historically, the potential application of JAK inhibition has been thoroughly explored, thus triggering an escalation of favorable results in this field. So far, five JAK inhibitors have been approved by the Food and Drug Administration (FDA) for the treatment of different diseases. Considering the complexity of JAK-depending processes and their involvement in multiple disorders, JAK inhibitors are the perfect candidates for drug repurposing and for the assessment of multitarget strategies. Herein we reviewed the recent progress concerning JAK inhibition, including the innovations provided by the release of JAKs crystal structures and the improvement of synthetic strategies aimed to simplify of the industrial scale-up.

## 1. Introduction

The pathogenesis of inflammatory states is an intricate complex of extracellular and intracellular mechanisms involving a considerable variety of enzymes, receptors and chemical mediators. The field of inflammation and pain remains one of the most investigated for new pharmacological approaches. Currently, nonsteroidal anti-inflammatory drugs (NSAIDs), corticosteroids and opiates rank as the most commercialized drugs [1,2].

Despite the debatable misuse of these medications for mild issues, their application in several pathological conditions represents a lifeline for some patients. In particular, chronic inflammatory states are still a significant challenge, and the study of inflammatory diseases often collides with the fields of autoimmunity and cancer [3,4,5,6].

The involvement of the same pathways in the pathogenesis of different diseases often offers the opportunity for off-label studies for drug repurposing, also providing the chance for multitarget approaches [7].

Janus kinases (JAKs) belong to the family of protein tyrosine kinases (PTKs). The JAKs subgroup consists of four members: JAK1, JAK2, JAK3 and Tyrosine Kinase 2 (TYK2). JAKs are stimulated by numerous cytokines (interleukins, interferons and hormones) and modulate the activation of signal transducer and activator of transcription (STAT) proteins by sitting at the top of signaling pathways that regulate several transcription processes (Figure 1) [8,9,10]. Cytokines are a group of polypeptide growth factors with low molecular weight (30 kDa) mainly involved in inflammatory processes, and they function by binding to the extracellular domains of different receptor superfamilies, leading to downstream signaling cascades [11]. They employ a crucial role in several physiological functions, especially in cell growth regulation and in both innate and adaptive immunity, as well as in human diseases [12,13]. 

JAK1 and JAK3 are involved in the signaling pathways of multiple cytokines (IL-2, 4, 7, 9, 15 and 21) associated with T cell development, activation and homeostasis [14]. The ubiquitously expressed JAK1 plays an essential role in the modulation of IL-6 and gp130 cytokine signaling and CD4+T cell expansion, differentiation and memory formation [15], whereas JAK3 is chiefly expressed in lymphoid tissues and appears to be a selective regulator of lymphoid development and function within the immune system [16,17,18]. Both are involved in the signaling pathways of multiple cytokines (IL-2, 4, 7, 9, 15 and 21) associated with T cell development, activation and homeostasis [14].

Recently, many of these signal-transduction pathways have been thoroughly investigated, and the role of JAKs has increased in prominence leading to increased interest from the pharmaceutical industry [19,20,21].

Since the approval of ruxolitinib for the treatment of rheumatoid arthritis (RA) in 2011, the research revolving around JAK inhibitors (JAKis) has come a long way (Figure 2). Ruxolitinib, and soon after Tofacitinib, had all the attention for a while. Despite having been approved for different disorders, their introduction placed the role of JAKs under a new light and made these kinases the object of growing interest. The increasing number of clinical trials focusing on JAKis has attracted the attention of the pharmaceutical companies, which initiated the need for more research in this area, and more efficient production strategies.

## 2. The Rise of JAK Inhibitors

In 1996, a compound with JAK2 inhibitory activity was reported to inhibit acute lymphoblastic leukemia (ALL) cell growth [22]: it was the first evidence of the role of JAKs as druggable targets. 

In 2011, fifteen years later, ruxolitinib was the first JAK inhibitor (JAKi) to be approved by the Food and Drug Administraion (FDA) (Figure 3). Ruxolitinib is a JAK1/2 selective inhibitor for the treatment of intermediate and high-risk myelofibrosis (MF) [23,24,25]. The brand name of the drug is Jakafi^®^, and it is distributed by Incyte Corp in the US. Recently, ruxolitinib has been evaluated in more clinical trials against several forms of cancer. A pilot study in 2019 suggested the possible role of ruxolitinib in the treatment of Hodgkin’s disease (HD) [26,27]. The effects of ruxolitinib in association with ibrutinib (an FDA approved Bruton’s tyrosine kinase (BTK) inhibitor) against chronic lymphocytic leukemia (CLL) were investigated in a phase 1 study and encouraging results were obtained [28]. In 2014, ruxolitinib was further approved for the treatment of polycythemia vera (PV) [29]. The drug was also recently evaluated in a phase 2 trial for its activity in graft-versus-host-disease (GvHD) [30,31].

Tofacitinib is a JAK1 and JAK3 inhibitor with a minor affinity for JAK2 and TYK2 [18] (Figure 3). Tofacitinib was developed in 2012 by Pfizer, and approved by the FDA under the name Xeljanz^®^ as a treatment against RA [32,33]. Recently, tofacitinib underwent Phase 3 clinical trials for the treatment of ulcerative colitis (UC) and was effective in improving the health-related quality of life of the patients [34,35]. Further studies demonstrated its efficacy against alopecia areata (AA) [36,37] as well as chronic plaque psoriasis (Ps) [38,39]. In 2017, tofacitinib was approved by the FDA to treat active psoriatic arthritis (PsA) [40].

Six years later, in 2018, Olumiant^®^ was approved by the FDA (Figure 3). Eli Lilly developed the drug, and the active ingredient was baricitinib, a JAK1 and JAK2 selective inhibitor for the treatment of RA [41]. Baricitinib also showed promising results in phase 2 clinical trials against moderate-to-severe atopic dermatitis (AD) [42], systemic lupus erythematosus (SLE) [43] and diabetic kidney disease (DKD) [44,45].

On August 16th, 2019, the FDA approved fedratinib (Inrebic^®^) and upadacitinib (Rinvoq^®^), two selective inhibitors of JAK2 and JAK1, respectively (Figure 3). Celgene Corporation developed fedratinib for the treatment of adult patients with intermediate-2 or high-risk primary or secondary post-PV MF [46]. Upadacitinib, distributed by AbbVie, is used in the treatment against RA [47,48]. A Phase 2/3 trial recently completed, also highlighted a possible role of upadacitinib in the treatment of active ankylosing spondylitis (AS) [49].

During the last few years, several additional new JAKis, which are under FDA evaluation, have entered phase 2 or phase 3 clinical trials (Table 1; Figure 3).

To date, JAKis hold their place among the best available therapies (BATs) for several forms of inflammatory diseases, in particular for those related to autoimmune disorders [61,62]. In the frontier against cancer, only one JAKi (ruxolitinib) has been approved so far; nevertheless, it seems clear that JAKs play a crucial role in numerous forms of cancer [63,64]. Preclinical data is available on this topic, and most of them are the product of genetic studies aimed to identify mutations responsible for pathological phenotypes. The JAK2 V617F mutation is probably the most discussed as it is strictly related to myeloproliferative pathologies [65,66]. 

Mutations in JAK1 and JAK3, as well as other rare mutations in JAK2, have been associated with T-cell acute lymphoblastic leukemia (T-ALL) [67,68]. Similarly, JAK1 mutations were found in 9% cases of hepatocellular carcinoma; in particular, the S703I mutation is likely responsible for the loss of JAK1 auto-inhibition ability [69,70,71].

The JAK1/2 blockade has also found a place among the considered strategies for the treatment of AIDS. Treatment with ruxolitinib showed significant results in controlling the levels of reservoir T-cells in HIV-1 patients [72,73].

In recent times, baricitinib entered several clinical trials for the treatment of coronavirus disease 2019 (COVID-19) pneumonia, providing promising results in terms of patient outcome [74,75,76]. The inhibition of JAKs is likely to play a key role in placating the cytokine storm responsible for the severe acute respiratory syndrome associated with COVID-19 [77].

These studies remain preliminary, but they still serve the purpose of reminding how much progress has been made in the field of JAKs since 1989 when Wilks claimed he had discovered “just another kinase” [64,78].

## 3. JAKis as Important Players in Multifactorial Diseases

The biological complexity of multifactorial diseases has led to a growing interest in polypharmacology and multitarget drug discovery. The multitarget approach is based on the modulation of multiple proteins rather than a single target to achieve a better therapeutic effect. In the last few years, this research field was considerably grown, thus offering new paradigms to overcome the limitations of the classic “a single key for a specific lock” strategies (Figure 3) [79,80]. Researchers aim to identify “master keys” able to recognize sets of “multiple locks”. The purpose is to give synergistic effects in multifactorial diseases such as cancer, inflammation and metabolic and neurodegenerative disorders [81,82,83].

Compared with the use of single-target drug combinations, the multitarget approach shows different advantages, such as higher efficacy, improved safety profile and simpler administration [84]. Drugs with multitarget profile effects can provide advanced therapeutic outcomes and a decrease in the side effect profile. This approach also facilitates the drug repositioning process hence promoting the use of approved and commercialized drugs for new therapeutic purposes (Figure 4) [85]. 

The multitarget potential of molecules is considered a brilliant emerging strategy for the treatment of diseases where multiple pathways are involved [86]. Indeed, inflammatory diseases like rheumatoid arthritis and autoimmune diseases are pathological conditions characterized by the activation and modulation of different pathways [15]. Recently, this approach was successfully applied for the repurposing of resveratrol, a natural compound that resulted in activity against serum/glucocorticoid-inducible kinase 1 (SGK1) [87].

Considering the role that JAKs play in these mechanisms, it has been proposed that the dual targeting of multiple JAKs may lead to enhanced cellular effects and potency and better immunosuppressive results [88].

Ruxolitinib is a JAK1 and JAK2 inhibitor with moderate inhibitory activity against TYK2 [89]. Ruxolitinib was developed for the treatment of PV and intermediate- and high-risk primary MF including primary MF, post-PV MF and post-essential thrombocythemia MF, where inappropriate activation of JAK2 underlies disease pathogenesis [90]. Chronic alterations in the JAK-STAT signaling have been identified in the pathogenesis of MF [91]. Cellular targets of ruxolitinib include both the innate and adaptive immune systems, such as natural killer cells, dendritic cells, T helper and regulatory T cells. Indeed, ruxolitinib exerts potent anti-inflammatory and immunosuppressive effects.

Immunomodulatory properties of ruxolitinib were also proven beneficial in corticosteroid refractory acute GvHD [92,93]. Ruxolitinib is also effective in treating of essential thrombocythemia and psoriasis [94,95], and in a recent study, it was repurposed as an anticancer agent for solid tumors [96] (Figure 5). In a recent clinical trial, ruxolitinib was also considered for drug-repurposing in the treatment of COVID-19 for its anti-inflammatory and immunosuppressive properties [97,98].

Treatment with ruxolitinib, due to its off-target effects, can cause hematologic adverse reactions, including thrombocytopenia, anemia and neutropenia and it can also increase the risk of developing serious bacterial, mycobacterial, fungal and viral infections [91]. 

Tofacitinib is a reversible and competitive antagonist that binds to the ATP-binding site in the catalytic cleft of the kinase domain of JAKs. Tofacitinib, specifically, inhibits JAK1, JAK2, JAK3 and to a lesser extent, TYK2, also presenting a high degree of selectivity against other kinases in the human kinome [99]. In the cellular environment, where JAKs signals are coupled, tofacitinib preferably inhibits the signaling by cytokine receptors associated with JAK3 and/or JAK1 with functional selectivity over JAK2 [100]. So far, tofacitinib is the only approved dual JAK1/3 inhibitor, which has demonstrated efficacy for autoimmune diseases and has been tested in the treatment of skin diseases [38,101,102]. Tofacitinib has also shown effectiveness in psoriasis, UC, kidney transplant rejection and Crohn’s disease (Figure 6). Additionally, tofacitinib was repurposed in the therapy against myeloma to reverse the growth-promoting effects of the bone marrow microenvironment [103].

It is interesting to mention that tofacitinib shows a moderate inhibitory activity against rho-associated protein kinase 2 (ROCK-2) and lymphocyte-specific protein tyrosine kinase (Lck) [104]. Although the affinity for these enzymes is thirty times smaller than the affinity for JAK1, these results may still be useful for the development of dual inhibitors (JAK/ROCK-2 or JAK/Lck) in the future.

This small molecule was approved despite its different off-target effects, such as the increased risk of developing severe infections, cancer, lymphoma or tuberculosis [103]. 

## 4. The Structure of Janus Kinases 

The JAK family includes four mammalian members, JAK1, JAK2, JAK3 and TYK2, characterized by a JAK homology pseudokinase (JH2) domain that regulates the adjacent protein kinase domain (JH1). Each JAK has specificity for a distinct set of cytokine receptors [105,106]. Cytokines drive JAK activation by binding to the extracellular domain of the receptor resulting in phosphorylation of JAK proteins that will serve as the docking site for STATs. Once activated by phosphorylation, STAT proteins dimerize and move to the nucleus where they regulate gene transcription (Figure 7) [9,12,107]. To our knowledge, more than forty cytokines rely on the JAK-STAT pathway, including IFN-γ, IL-2, IL-6, IL-12 and IL-23 [107].

The structural architecture of JAKs has been the object of extensive examination through the years (Figure 8A) [106,108,109,110]. Nevertheless, a few key points can capture the essence of these efforts and outline their main features:(a)JAK kinases are large, multidomain proteins (120–130 kDa) initially organized into seven different JAK homology (JH) domains;(b)Four domains are currently considered the most relevant: the *N*-terminal four-point-one, ezrin, radixin, moesin (FERM) domain (JH5/6/7), the Src homology 2 (SH2) or SH2-like (SH2L) domain (JH3/4), the pseudokinase domain (Ps-JH2) and the *C*-terminal tyrosine kinase domain (JH1);(c)The FERM and SH2 domains form a unique structural assemblage (Figure 8B);(d)JH1 represents the catalytic kinase domain (Figure 8C);(e)JH2 might negatively regulate JAK2 function by the phosphorylation of key residues within the JH2 domain;(f)JH3-JH4 stabilize the conformation of the JAKs;(g)JH5-JH7 directly interact with the intracellular domains of the cytokine receptor and with the JH1 domain.

JAKs contain a small *N*-terminal lobe and a large *C*-terminal lobe, including several conserved residues. A twisted antiparallel β-sheet and a regulatory α-C-helix are present in the small lobe, above the ATP binding site (Figure 8D) [111]. Relevant residues are K908 (JAK1 K908, PDB: 3EYG) on the β3-strand and E925 on the α-C-helix (JAK1 E925, PDB: 3EYG). On the α-helical *C*-terminal lobe two aspartate residues are located (JAK1 D1003, D1021-PDB: 3EYG) that form the activation segment of JAK1. Finally, E957 (E957 of JAK1 JH1 domain) and L959 (JAK1 L959, PDB: 3EYG) are involved in the interactions with steady-state ATP competitive inhibitors (Figure 8D).

Tofacitinib is a first-generation JAK inhibitor, a competitive antagonist that binds to the ATP-binding site (Figure 9A). Tofacitinib inhibits the phosphorylation and activation of JAK, thereby avoiding the phosphorylation and activation of STATs. Consequently, gene transcription is suppressed and the result of these events is a significant reduction of cytokine production and a modulation of the immune response is observed [112,113]. Its interactions within the ATP binding site play a key role in defining the structure–activity relationship (SAR) profile of the drug. N7 of the pyrrolo [2,3-d]pyrimidine scaffold makes one hydrogen bond with the carbonyl group of E957, and N1 of the pyrimidine ring forms a second hydrogen bond with the amino group of L959 in JAK1 (Figure 9B–E). The cyanoacetyl moiety significantly contributes to the stabilization of the structure as it extends beneath the β-3-sheet and in front of K908 on the α-helical C-terminal lobe. It is worth noticing how the methyl group of the piperidine ring was exposed to the solvent in the front cleft (Figure 8D; Figure 9B,D,E) and many hydrophobic contacts with L881, V889, A906, V938, M956, P960 and L1010 were detected (Figure 10).

Among the five FDA approved JAKis, tofacitinib was the only one to be cocrystallized with a JAK isoform so far. The definition of its binding pose provided essential information and an excellent tool for the design of new inhibitors. Due to the advances in the fields of computational chemistry and bioinformatics, the release of a protein–inhibitor complex can be exploited for the application of in-silico techniques such as high-throughput virtual screening (HTVS) or pharmacophore design. These approaches allow the shortening of the development time of new drugs, and they often are considerably cheaper alternatives, mostly in the early stages of the drug-discovery process.

## 5. Synthetic Strategies for the Development of Ruxolitinib and Tofacitinib 

For six years, ruxolitinib and tofacitinib were the only JAKis available on the market. Their synthetic strategies have been revised several times through the years to allow a smoother transition to the industrial scale-up and balance out the equilibrium among sustainability, production costs and actual benefits for the producers.

The JAKis that have been investigated so far more or less maintain several similarities in their scaffold, hence the role played by the research of multiple strategies for the synthesis of the “older generation of JAKis” must not be underestimated. As the approval of the first two JAKis represents a milestone, the development of more advanced synthetic strategies is the first step towards the future.

### 5.1. The Synthesis of Ruxolitinib

Ruxolitinib is a small molecule composed of a substituted pyrazole linked to a deazapurine core (Figure 11).

The substituted pyrazole ring is the key compound for the synthesis of ruxolitinib. Despite being the first therapeutic JAK1 and JAK2 inhibitor, only two articles have been published [114,115]. Recently, a patent was registered that provides an alternative procedure to obtain ruxolitinib on an industrial scale [116].

According to the few information available, the substituted pyrazole ring was obtained using different procedures and starting from several commercially available materials (Figure 12). 

In 2009 Lin et al. [114] synthesized ruxolitinib by an asymmetric organocatalytic Aza-Michael reaction, which encompasses the addition of a substituted pyrazole to an α,β-unsaturated aldehyde. The authors provided two different approaches to obtain the substituted pyrazole moiety, which is finally linked to the deazapurine synthon to afford ruxolitinib (Scheme 1A,B). In Scheme 1A, 4-iodopyrazole is used to provide the appropriate *N*-substituted pyrazole unit as a Michael donor. Alternatively, 4-bromopyrazole is directly treated with an α,β-unsaturated aldehyde to yield ruxolitinib (Scheme 1B). Following the first strategy, the reaction of 4-iodopyrazole with ethyl vinyl ether provided the protected pyrazole, which was treated with methoxy pinacol borate to yield the pyrazole pinacol borate (Scheme 1A, steps a and b). Then, Suzuki coupling of the latter with TMS-protected-6-chloro-7-deazapurine (Scheme 1A, step c) and further hydrolysis of the *N*-protecting group furnished the key *N*-substituted pyrazole (Scheme 1A, step d). An asymmetric aza-Michael reaction was performed between this derivative and *E*-3-cyclopentyl-acrylaldehyde, using an appropriate organ-catalyst to yield the corresponding aldehyde. (Scheme 1A, step e). Finally, the latter was treated with aqueous ammonia and iodine to give the corresponding nitrile (Scheme 1A, step f). The TMS protecting group was removed using lithium tetrafluoroborate (LiBF_4_) to afford ruxolitinib.

The same research group reported an alternative approach for the synthesis of ruxolitinib, exploiting the aza-Michael reaction of 4-bromopyrazole with *E*-3-cyclopentyl-acrylaldehyde (Scheme 1B, step a) [114]. The attained aldehyde product was converted into the nitrile derivative using NH_4_OH/I_2_ (Scheme 1B, step b), which in turn provide ruxolitinib by a Suzuki coupling of the in situ generated pinacol borate with the 6-chloro-7-deazapurine derivative (Scheme 1B, steps c and d).

Haydl and coworkers [115] further investigated the synthesis of the *N*-substituted pyrazole derivative and the role of this compound as a ruxolitinib precursor (Scheme 1C). In particular, the appropriate *N*-substituted pyrazole was synthesized by a rhodium-catalyzed asymmetric addition of commercial pyrazole to allene. Then, inspired by this synthetic possibility, this new pyrazole allylation was applied to the synthesis of ruxolitinib. The rhodium-catalyzed coupling of 4-bromopyrazole with the cyclohexylallene provided the corresponding alkene derivative (Scheme 1C, step a), which was submitted to hydroboration and oxidation to yield the corresponding aldehyde (Scheme 1C, steps b and c). Finally, as observed for the previously reported procedures, ruxolitinib was obtained by a Suzuki reaction (Scheme 1C, steps d–f).

As described in Scheme 1, ruxolitinib essentially was obtained by linking the deazapurine core with the *N*-substituted pyrazole through a Suzuki reaction. However, the reactions performed to synthesize the pyrazole moiety require high amounts of expensive and complex chiral catalysts [114,115]. Furthermore, the regio- and enantioselectivity of the product obtained by chiral resolution is not high, and the product usually undergoes a multistep refining process to achieve the final compound [114]. Compared to the methodologies proposed by Haydl [115], the amounts of used rhodium catalyst and metal ligand are relatively high and the reaction conditions are quite tough. This method is expensive and is not applicable for industrial production [115]. Therefore, to overcome such hurdles, a patent was recently published providing a new method for the preparation of ruxolitinib, which was suitable for industrial production [116], as outlined in Scheme 2. The cyclopentanecarbaldehyde was selected as starting material to provide the N-substituted pyrazole. In particular, the aldehyde was converted into the correspondent cyclopentyl-pyrazolidinone using malonic acid and hydrazine (Scheme 2, steps a and b). The racemic mixture was resolved with di-p-toluoyl-L-tartaric acid (L-DTTA; Scheme 2, step c). On the other hand, the 6-methyl-7-deazapurine derivative was treated with the Vilsmeier reagent (POCl_3_/DMF) to provide the corresponding aldehyde product (Scheme 2, step d). This deazapurine derivative reacted with the pure enantiomer cyclopentyl-pyrazolidinone in alkaline condition (NaOH) to yield the *N*-substituted-deazapurine pyrazole (Scheme 2, step e). The carboxylic group was activated with oxalyl chloride (Scheme 2, step f) and treated with NH_3_ (Scheme 2, step g) and POCl_3_/*N*-Methylpyrrolidone (NMP; Scheme 2, step h) to provide ruxolitinib. Following this approach, the JAKi was obtained in good yield, avoiding the use of complex and expensive catalysts [116].

### 5.2. The Synthesis of Tofacitinib

Tofacitinib can be fragmented in two synthons, the (3*R*,4*R*)-disubstituted piperidine and the deazapurine core (Figure 13).

Depending on the chosen synthetic strategy, the synthesis of the (3*R*,4*R*)-disubstituted piperidine derivative can be achieved from different commercially available starting materials (Figure 14). The preparation of the piperidine moiety is challenging and requires a final chiral resolution step due to the presence of two asymmetric centers. Therefore, several synthetic protocols have been published to obtain the key (3*R*,4*R*)-disubstituted piperidine derivative [117,118,119,120,121].

Pfizer Inc. [117,118] reported the first synthetic strategies for the synthesis of tofacitinib. As outlined in Scheme 3, the key compound (3*R*,4*R*)-1-benzyl-N,4-dimethylpiperidin-3-amine was obtained from two different starting materials and following two protocols.

The first approach [117] (Scheme 3A) encompasses the reaction of 4-methylpicoline with benzyl chloride, followed by a selective reduction, providing the *N*-benzyl tetrahydropiperidine (Scheme 3A, steps a and b), which in turn was treated with BF_3_ OEt_2_, BH_3_ and H_2_O_2_ to give the alcohol derivative 1-benzyl-*N*,4-dimethylpiperidin-3-amine (Scheme 3A, step c). This derivative was first submitted to oxidation, and then to reductive amination, followed by a chiral resolution step with L-DTTA, which afforded the (3*R*,4*R*)-1-benzyl-*N*,4-dimethylpiperidin-3-amine (Scheme 3A, steps c–e). Although the key compound was isolated, some drawbacks limited its application for industrial production, including the use of BF_3_, for the release of toxic hydrogen fluoride and the sulphur-based oxidation, being the dimethyl sulphide a contaminant of the final product. Therefore, the same company proposed an alternative synthetic strategy to avoid the aforementioned problems.

Following Scheme 3B [118], the key synthon was achieved by condensation of 3-amino-4-methylpyridine with dimethyl carbonate (Scheme 3B, step a), followed by the hydrogenation of the pyridine ring (Scheme 3B, step b) and the benzyl-protection of the *sec* amine group of the piperidine moiety (Scheme 3B, step c). Finally, the carbamate group was hydrolyzed (Scheme 3B, step d) and the 3*R*, 4*R* enantiomer was obtained by resolution with L-DTTA. Although this route appeared reasonable, it was still affected by different problems such as the use of inflammable and explosive LiAlH_4_.

Nevertheless, tofacitinib was synthesized by nucleophilic substitution of the (3*R*,4*R*)-1-benzyl-*N*,4-dimethylpiperidin-3-amine with 6-chloro-7-deazapurine in alkaline condition (Scheme 3, step f), and subsequent easy deprotection and amidation reactions [117,118] (Scheme 3, steps g and h).

Due to the pharmaceutical importance, and to overcome the synthetic constraints reported by Pfizer Inc., the synthesis of tofacitinib was extensively studied and several procedures were published [119,120,121,122,123,124,125]. Usually, the key compound implicated in the synthesis of tofacitinib is the (3*R*,4*R*)-1-benzyl-*N*,4-dimethylpiperidin-3 amine, which is obtained from 3-amino-4-methylpyridine (Scheme 4A,B) [119,120] or 4-methyl-picoline (Scheme 4C). Alternatively, 4-methyl-picoline was also used to synthetize 1-benzyl-3-hydroxy-4-methylpiperidine, as an intermediate to provide tofacitinib (Scheme 4D) [121].

According to Patil and coworkers [119], (3*R*,4*R*)-1-benzyl-*N*,4-dimethylpiperidin-3 amine was obtained in two steps without the use of any expensive reagent (Scheme 4A). The chemistry involves the *N*-acylation and *N*-benzylation of the starting material using acetyl and benzyl chloride (Scheme 4A, steps a and b) and the subsequent selective reduction with NaBH_4_ (Scheme 4A, step c) to provide the isolated tetrahydropiperidine intermediate. The synthon was obtained through the hydrolysis of the amide group (Scheme 4A, step d) and the subsequent reductive amination using methylamine and NaBH_4_ (Scheme 4A, step e). The resolving agent L-DTTA was used to yield the pure enantiomer (Scheme 4A, step f).

Zhi et al. [120] reported an economical, efficient and improved procedure for the synthesis of tofacitinib (Scheme 4B). This method was proposed as an improved alternative compared to the Pfizer approach [118]. In particular, the carbalkoxylation (Scheme 4B, step a) was performed using NaH, the *N*-benzyl group was linked through nucleophilic substitution (Scheme 4B, step c), and finally the amide group was hydrolyzed with sodium dihydro-bis-(2-methoxyethoxy) aluminate (Red/Al; Scheme 4B, step d). The optimization of the reaction steps and purification processes, such as the selection of the catalysts, the benzyl protection and the reductive agents, obviously improved the yield of tofacitinib approximately to 13.3%, compared with 8.6% [118,126].

More recently, Srishylam et al. [121] reported a completely new approach for the synthesis of Tofacitinib (Scheme 4C,D). According to the authors, the key step of this synthesis was the formation and purification of the *N*-benzyl-piperidine epoxide, which was finally achieved by using m-chloro-peroxybenzoic acid (m-CPBA; Scheme 4C, step c). The regioselective ring-opening with sodium azide (Scheme 4C, step d) and the further reduction of the *N*-Boc protected aziridine using LiAlH_4_ (Scheme 4C, step f) provided the 3-amine-4-methyl-*N*-benzyl piperidine. The resolution of the racemic mixture with L-DTTA yielded the 3*R*-4*R* enantiomer (Scheme 4C, step f).

Tofacitinib was easily synthesized through a nucleophilic substitution between the key synthon and the commercially available 4-chloropyrrolo[2,3-d]pyrimidine tosilate (Scheme 4, step g), the removal of the *N*-tosyl (Scheme 4, step g) and *N*-benzyl protecting group (Scheme 4, step h) and an amidation reaction (Scheme 4, step i). In general, the nucleophilic substitution was carried out in alkaline conditions (K_2_CO_3_) [119,120,121], the tosyl and benzyl groups were removed using KOH and Pd/C [119] or NaOH and Pd(OH)_2_ [120]. The amidation reaction was performed with ethyl 2-cyanoacetate and DBU as a catalyst agent [119,120,121]. As an alternative to the nucleophilic substitution approach, the tofacitinib precursor can be prepared by a Mitsunobu reaction with 6-methylamino-7-deazapurine and 3-hydroxy-4-methyl-*N*-benzyl piperidine (Scheme 4D, step b).

As reported above, (3*R*,4*R*)-1-benzyl-*N*,4-dimethylpiperidin-3 amine is considered as the key compound for the synthesis of tofacitinib. 

In general, to obtain the pure enantiomer, the racemic mixture is resolved with L-DTTA or other resolving agents, increasing the cost and time of synthesis. To overcome this non-negligible issue, an asymmetrical synthesis of tofacitinib, which avoids the chiral resolution step, was developed [122,123].

In this context, Marican et al. [122] proposed an alternative strategy based on the precursor tert-butyl-(3*S*,4*R*)-3-hydroxy-4-methylpiperidine-1-carboxylate (Scheme 5A). Starting from (5*S*)-5-hydroxypiperidinone, the key synthon was obtained by subsequent reactions, which mainly consisted of the protection of the alcohol and amide groups using tert-butyl(chloro)diphenylsilane (TBDPSCl) and Boc anhydride respectively (Scheme 5A, steps a, b), oxidation with hexamethyldisilazane (HMDS) and H_2_O_2_ (Scheme 5A, step c), and then the Grignard reaction using MeMgBr/CuBrSMe_3_ (Scheme 5A, step d). Finally, the removal of the TBDPSCl alcohol protecting group yielded the tert-butyl-(3*S*,4*R*)-3-hydroxy-4-methylpiperidine-1-carboxylate (Scheme 5A, step e). Tofacitinib was then obtained by a Mitsunobu reaction using 6-methylamino-7-deazapurine and triphenylphosphine/diisopropyl azodicarboxylate (Ph_3_P/DIAD) as a catalyst (Scheme 5A, step f), the Boc deprotection in the presence of ZnBr_2_ (Scheme 5A, step g), and a *N*-(3-dimethylaminopropyl)-*N*′-ethylcarbodiimide/hydroxybenzotriazole (EDC/HOBt)-coupled amidation reaction with cyanoacetic acid (Scheme 5A, step h).

In line with this purpose, Liao and coworkers reported another asymmetric synthesis for tofacitinib. The key step, in this case, was the synthesis of the (3*R*,4*R*)-1-benzyl-4-methyl-3-hydroxypiperidine intermediate (Scheme 5B). The main steps of this procedure encompassed the synthesis of the γ-lactone (Scheme 5B, step b), the expansion of the ring and formation of the diamides using LDA (Scheme 5B, step c), and the reduction of the corresponding intermediate with LiAlH_4_ to provide the key compound (3*R*,4*R*)-1-benzyl-4-methyl-3-hydroxypiperidine (Scheme 5B, step d). Tofacitinib was obtained by nucleophilic substitution (Scheme 5B, step e), and an alternation of *N*-protecting group removal and amidation reaction as previously reported [119].

Recently, the development of protocols describing the conditions for catalytic asymmetric reductive amination for the preparation of chiral amines as key intermediates in the synthesis of tofacitinib was also documented [124,125].

Considering the pharmacological importance of both JAKis, the syntheses of ruxolitinib and tofacitinib were exhaustively studied. Compared with the first syntheses published for both compounds, important progresses have been made and the selection of appropriate reagents, especially catalyst agents, and reaction conditions provided sustainable and innovative synthetic procedures fitting with industrial production.

## 6. Enhancing JAKis Discovery through Structure-Based Drug Design Strategies 

The approval of new drugs averagely requires time and huge investments of money, enough that in recent years a 100% increase in drug development costs has been estimated [127].

Currently, the innovations in IT offer a plethora of new resources in the scientific field, including an ample range of cheminformatic and bioinformatic tools for drug discovery. Consequently, the possibility to perform hundreds of simulations simultaneously and in the shortest time represents the key feature of computer-aided drug design (CADD). Noteworthy, CADD is not only a peculiarity of academic research, but it is a well-established practice even at the industrial level. At the same time, plenty of technologies for structure determination (X-ray crystallography and NMR), have progressed, giving rise to the new era of structure-based drug design (SBDD) in drug discovery. From crystallographic data, SBDD can reveal the ligands’ binding mode in the target pocket, hence providing one of the most favorable ways for the design of potential high-affinity ligands. SBDD, in addition to being preferred by drug designers because of its higher success rate, has emerged as the most innovative tool used in the pharmaceutical industry helping researchers to individuate many lead compounds in different research fields [128,129,130,131,132,133,134].

Recently, the release of the crystal structures of the JAK isoforms allowed the SBDD application for the development of JAKis. Several examples are available, which provided encouraging results and future perspectives for the discovery of new molecular entities targeting JAKs. In 2016, a study describing the structure-based design of (benz)imidazole pyridones was published. The peculiarity of this work consisted in exploiting the available crystallographic information to develop JAK1-selective and higher affinity inhibitors. The outcome not only afforded a *hit* compound with high specificity towards JAK1, but also highlighted the importance of targeting key residues within the binding site of JAK1 (Arg879 and Glu966) [135].

Other relevant results concerning JAK1 selectivity were obtained from a fragment-based approach carried out by Hansen et al. [136]. The work was actually a follow-up of a previous one in which the design of 6-arylindazole derivatives had been performed using a joint fragment-based and structure-based approach. Unfortunately, although able to significantly inhibit all the JAKs isoforms, all the evaluated compounds were unstable and phototoxic [137]. In the 2020 work, the team started from a fragment previously evaluated, which showed enhanced JAK1 affinity compared to JAK2. The hit expansion of the fragment allowed the development of several molecular entities with improved affinity towards JAK1. Eventually, due to the information provided by the crystal structure, new molecular entities were obtained showing optimized features and a better interaction with the lipophilic pocket and the p-loop. The main outcome of the study was indeed the development of a pyrazolopyridone inhibitor of JAK1 with high selectivity and nM efficacy [136]. 

A recent series of three works concerning the design and development of JAK3 selective inhibitors resulted from the joined efforts of two Chinese teams. The articles explore the design and synthesis of 1H-pyrazolo[3,4-d]pyrimidin-4-amino derivatives [138], pyrimidine-4,6-diamine derivatives [139] and 4- or 6-phenyl- pyrimidine derivatives [140], respectively. The three studies share a common pattern consisting in the design of derivatives, in vitro evaluation and a docking analysis to better understand the interaction framework and explain the selectivity. Overall, among the examined compounds, two eventually demonstrated great affinity for JAK3 and showed high selectivity with respect to the other isoforms [139,140]. Interestingly, the docking studies pointed at Cys909 as a key residue for JAK3 selectivity, confirming a previous hypothesis according to which the aryl group of the designed derivative forms a covalent bond with the Cys909 sulphur group [139,140].

Further interesting studies concerning JAK selectivity were carried out by Bajusz et al. starting in 2016 and are still ongoing [141,142]. The research group succeeded in developing a JAK2 selective compound with nM affinity. The initial efforts of the 2016 study consisted of the development and validation of a structure-based in silico protocol able to discriminate between an active and a decoy dataset. The method was further implemented to recognize possible *hits* with JAK2 over JAK1 selectivity. The main result was the discovery of an indazole derivative with a 14-fold JAK2/JAK1 selectivity and 2 μM affinity towards JAK2 [141]. The same compound was hence considered for optimization in the next work. A series of derivatives was designed and synthesized and, eventually, 3-Amino-N-(2,6-dichlorophenyl)-1H-indazole-5-carboxamide was isolated as the most promising *hit* showing nM affinity towards JAK2 and 16-fold JAK2/JAK1 selectivity. Noteworthy, these computational studies also revealed that the water molecules network within the binding site plays a pivotal role in the formation of ligand–protein interactions and must be hence taken into account when performing CADD [142].

## 7. Conclusions

In 2011, ruxolitinib was approved as the first JAK inhibitor for clinical use. That first result seems so distant now, considering all the progress achieved in this field and the considerable efforts made to discover new molecular entities able to inhibit JAKs. Up to date, also tofacitinib, baricitinib, fedratinib and upadacitinib have obtained FDA approval as JAKis, and several other compounds have reached the final stage of clinical trials. Lately, the involvement of JAKs in several complex diseases, including cancer, inflammation and autoimmune diseases, was determined. In this context, the search of new therapeutic agents as JAKis became a hot topic. These findings offered new perspectives to fight autoimmune-related diseases and severe inflammatory disorders.

This review was organized to give an outline of the last four years’ progress in the development of JAKis. The multitarget spectrum of the “older generation” of approved JAKis, ruxolitinib and tofacitinib, was also discussed, given the prospects opened by this kind of approach. Nevertheless, the main focus of this review covered the aspects related to the synthesis of ruxolitinib and tofacitinib, thus offering an analysis from the original to the most innovative and interesting approaches for the production of these compounds.

Ruxolitinib and tofacitinib remained the only JAKis approved in therapy for at least six years. Furthermore, most of the recent JAKis share the deazapurine core present in the ruxolitinib and tofacitinib scaffold. Therefore, the synthesis of the “older generation of JAKis” must not be neglected. In particular, concerning the synthesis of ruxolitinib, only two articles and a more recent patent have been published. The patent offers a solution to some of the synthetic issues showed by the older synthetic methodologies and represents a cost-effectiveness and sustainable strategy for the industrial production of ruxolitinib. On the other hand, the synthesis of tofacitinib has been explored more and several synthetic procedures have been proposed. The strategies mainly differ in the preparation of the piperidine moiety, a challenging step due to the presence of two chiral centers. The first methodologies reported by Pfizer, Inc. were improved during the years, providing economically efficient and sustainable procedures more fitting with the industrial production.

Innovative technologies are the driving force of drug discovery. The recent progress in the field of computational chemistry and bioinformatics has facilitated the research protocols, providing quicker and cheaper approaches to the preliminary evaluation of new active compounds. 

Computational techniques represent powerful and useful tools to reduce the time and to enhance rational drug design in the medicinal chemistry field. The combination of in silico approaches with the advances in purification and structure investigation techniques has resulted in reduced time and costs of drug development. Recent works offered a clear prospect of the benefits of SBDD for the identification of new JAK inhibitors, providing remarkable results.

In the future, the recent advances in the design and synthesis of JAKis will certainly boost the development process of new treatments offering alternative perspectives against complex diseases.

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
