# Peer review of "Inside Perspective of the Synthetic and Computational Toolbox of JAK Inhibitors: Recent Updates"

_molecules, 2020, doi:10.3390/molecules25153321_

Round 1

Reviewer 1 Report

The manuscript molecules-855842 “The Potential of JAK Inhibitors in the Treatment of Complex Diseases” attempts to describe the interactions between inflammation and cancer (complex diseases) and the use of JAK compounds. The review is very thorough with regard to the synthesis of novel JAK inhibitors using currently knowledge and synthesizing multiple derivatives of an active molecule backbone. The authors state that this review will examine the “recent progress concerning JAK inhibition, including the innovations provided by the release of JAKs crystal structures and the improvement of synthetic strategies aimed at the simplification of the industrial scale-up.”

Overall a solid submission but it just feels like the review misses the point.  From the title, there is an expectation that much of the review will be discussing current as well as new/novel JAK inhibitors in the treatment of complex disease processes.  Yet, the majority of the review discusses and outlines the synthesis of various JAK inhibitor analogs.  Essentially, from line 172 all the way to the end at line 495, the entire discussion is of the synthesis processes for different classes of JAK inhibitors and there is only a superficial discussion on the actual role of JAK (and its various isoforms), STAT and other mediators of the inflammatory process.  The following would be suggestions to improve the quality of the submission, or at least bring it more into line with the title.

MAJOR:

The title would need to be changed to be more reflective of what the review is covering (or visa versa, the text needs to be changed)

The beginning – from lines 1 to 172, there is good material in here and this section should be significantly expanded.

Conclusions?  There really isn’t any conclusion section.  This is where the current and the future should come together with in-depth speculation as to the current utilization of JAK inhibitors and then a deep dive into where JAK inhibitor use may be in the future.

MINOR:

16           change ‘Over the last decades,’ to ‘Historically,’

17           change the wording on ‘the possibilities of JAK inhibition have been thoroughly explored’ – this is a little awkward and hard to really determine – WHAT possibilities are being explored

19-20     change the sentence ‘Considering the complexity of JAK-depending processes and their involvement in multiple disorders, JAK inhibitors are the perfect candidates for drug repurposing and for the assessment of multi-target strategies.’ To ‘JAK inhibitors are the perfect candidates for drug repurposing considering the complexity of JAK-dependent processes and their involvement in multiple disorders.’

23           change ‘the simplification’ to ‘simplify’

57           change ‘During the past years,’ to ‘Recently,’ Phrases like ‘during the past years’ or ‘nowadays’ don’t always fit within the context of the writing style

58-59    the sentence fragment ‘and the role of JAKs has gradually shifted into the spotlight, hence attracting increasing interest from the pharmaceutical industry [19-21].’ Is wordy and because of that, can be hard to follow.  Consider ‘and the role of JAKs has increased in prominence leading to increased interest from the pharmaceutical industry [19-21].’

66-69    this section should be re-written.  Adding words/phrases like ‘tip of the iceberg’, ‘kicked off the exploration’, etc. could be streamlined and simplified.  For example, instead of:

‘The increasing number of clinical trials focused on JAKis was just the tip of the iceberg; at some point, the attention of the pharmaceutical companies naturally turned towards the necessity for more efficient production strategies while researchers everywhere kicked off the exploration of this field from broader perspectives.’

Can be shortened to:

‘The increasing number of clinical trials focusing on JAKis has attracted the attention of the pharmaceutical companies, which initiated the need for more research in this area, and more efficient production strategies.’

79           change ‘proved’ to ‘suggested’

88           change ‘resulted’ to ‘was’

91           change ‘for the treatment of’ to ‘to treat’

113         change ‘Several preclinical data are available on…’ to ‘Preclinical data is available on…’

132         delete ‘the fields of’

135         change ‘fields have’ to ‘field was’

137        delete ‘nowadays’ and change ‘at the identification of’ to ‘to identify’

140         indent the first line and then readjust lines 142-144 which appear to be a hanging indent.  Change ‘When compared with…’ to ‘Compared with…’

147         delete ‘in particular’

151         change ‘which’ to ‘that’ and change ‘active’ to ‘in activity’

161         delete ‘various components’ and change ‘system’ to ‘systems’

165         fix indent and then change ‘the treatment of’ to ‘treating’

180         delete the second ‘which’

183         delete ‘in a recent work,’

195-97   the sentence ‘By binding to the extracellular domain of the receptor, cytokines drive the activation by phosphorylation of JAK proteins that serve as the docking site for STATs.’ Should be rewritten.  Maybe to ‘Cytokines drive JAK activation by binding to the extracellular domain of the receptor resulting in phosphorylation of JAK proteins that will serve as the docking site for STATs.’

216         change ‘both of which include’ to ‘including’

236         the end of the sentence ‘STATs and consequently the gene transcription.’ Should be separated into 2 sentences, the ‘consequently gene transcription’ could be expanded.

237         change ‘The result of these events is a significant…’ to ‘Overall, a significant…’

238         insert ‘is observed’ after ‘response’

259         delete ‘nowadays’

267         delete the second ‘to’

278-80   the sentence ‘Despite being the first JAK1 and JAK2 inhibitor introduced in therapy, as far as our knowledge goes, there are only two articles [114,115], and a more recent patent [116] which resume the state of art and provides an alternative procedure to obtain Ruxolitinib in industrial scale.’ Is very long and can be separated into at least 2 sentences.  For example: ‘Despite being the first therapeutic JAK1 and JAK2 inhibitor, only two articles have been published [114,115]. Recently, a patent [116] which provides an alternative procedure to obtain Ruxolitinib on an industrial scale.’

293         delete ‘in turn’

308         what does ‘application to provide Ruxolitinib’ mean?  Seems like there is something missing there.

330-31   change ‘for the synthesis of’ to ‘to synthesize’

333         delete ‘most of the time’ and change ‘subjected to’ to ‘usually’

337         change ‘which provided’ to ‘providing’

338         change ‘a new method for the preparation of Ruxolitinib which was suitable for industrial’ to ‘a new method for the preparation of Ruxolitinib, which was suitable for industrial’

349         I believe ‘oxalylchloride’ is misspelled.

390        delete ‘and in order’

428         change ‘the synthesis of’ to ‘synthesizing’

433         change ‘process’ to ‘processes’

444         change ‘commercial’ to ‘commercially’

449         insert ‘a’ before ‘catalyst’

455         the sentence ‘…resolving agents, increasing the costs and time-consuming among other.’ Can be reworded for clarity to ‘…resolving agents, increasing the cost and time of synthesis.’

461         change ‘in’ to ‘of’

468         insert ‘the’ before ‘presence’

490         change ‘were’ to ‘was’

492-93   the sentence ‘According to the literature data, compared with the first syntheses published for both compounds, several synthetic constraints have been faced.’ Is unclear – this should be rewritten and maybe expanded a little for the improvement in clarity.

495         delete ‘the’

Author Response

Answers to Reviewer 1

The manuscript molecules-855842 “The Potential of JAK Inhibitors in the Treatment of Complex Diseases” attempts to describe the interactions between inflammation and cancer (complex diseases) and the use of JAK compounds. The review is very thorough with regard to the synthesis of novel JAK inhibitors using currently knowledge and synthesizing multiple derivatives of an active molecule backbone. The authors state that this review will examine the “recent progress concerning JAK inhibition, including the innovations provided by the release of JAKs crystal structures and the improvement of synthetic strategies aimed at the simplification of the industrial scale-up.”

Overall a solid submission but it just feels like the review misses the point.  From the title, there is an expectation that much of the review will be discussing current as well as new/novel JAK inhibitors in the treatment of complex disease processes.  Yet, the majority of the review discusses and outlines the synthesis of various JAK inhibitor analogs.  Essentially, from line 172 all the way to the end at line 495, the entire discussion is of the synthesis processes for different classes of JAK inhibitors and there is only a superficial discussion on the actual role of JAK (and its various isoforms), STAT and other mediators of the inflammatory process.  The following would be suggestions to improve the quality of the submission, or at least bring it more into line with the title.

Major Points

The title would need to be changed to be more reflective of what the review is covering (or visa versa, the text needs to be changed)

The title was changed according to the reviewer’s suggestions: Inside perspective of the synthetic and computational toolbox of JAK inhibitors: recent updates.

The beginning – from lines 1 to 172, there is good material in here and this section should be significantly expanded

A new section concerning the structure-based drug design of new JAKis was inserted. By doing this, and by changing the title, we think the review should now be clearer and more focused on innovations provided by the development of new methods and new technologies.

“6. Enhancing JAKis discovery through Structure-Based Drug Design Strategies

The approval of new drugs averagely requires time and huge investments of money, enough that in recent years a 100% increase in drug development costs has been estimated [127].

Nowadays, the innovations in IT afford a plethora of new resources in the scientific field, including an ample range of cheminformatic and bioinformatic tools for drug discovery. Consequently, the possibility to perform hundreds of simulations simultaneously and in the shortest time represents the key feature of computer-aided drug design (CADD). Noteworthy, CADD is not only a peculiarity of academic research, but it is a well-established practice even at the industrial level. At the same time, plenty of technologies for structure determination (X-ray crystallography and nuclear magnetic resonance, NMR), have progressed, giving rise to the new era of Structure-Based Drug Design (SBDD) in drug discovery. From crystallographic data SBDD can reveal the ligands binding mode in the target pocket, hence providing one of the most favourable ways for the design of potential high-affinity ligands. SBDD, in addition to being preferred by drug designers because of its higher success rate, has emerged as the most innovative tool used in the pharmaceutical industry, has helped researchers to individuate many lead compounds in different research fields [128-134].

Recently, the release of the crystal structures of the JAK isoforms allowed the SBDD application for the development of JAKis. Several examples are available which provided encouraging results and future perspectives for the discovery of new molecular entities targeting JAKs. In 2016, a study describing the structure-based design of (benz)imidazole pyridones was published. The peculiarity of this work consisted in exploiting the available crystallographic information to develop JAK1-selective and higher affinity inhibitors. The outcome not only afforded a hitcompound with high specificity towards JAK1, but also highlighted the importance of targeting key residues within the binding site of JAK1 (Arg879 and Glu966) [135].

Other relevant results concerning JAK1 selectivity were obtained from a fragment-based approach carried out by Hansen et al. [136]. The work was actually a follow-up of a previous one in which the design of 6-arylindazole derivatives was performed using a joint fragment-based and structure-based approach. Unfortunately, although able to significantly inhibit all the JAKs isoforms, all the evaluated compounds were unstable and phototoxic [137]. In the 2020 work, the team started from a fragment evaluated during the previous which showed enhanced JAK1 affinity compared to JAK2. The hit expansion of the fragment allowed the development of several molecular entities with improved affinity towards JAK1. Eventually, thanks to the information provided by the crystal structure, new molecular entities were obtained showing optimised features for a better interaction with the lipophilic pocket and the p-loop. The main outcome of the study was indeed the development of a pyrazolopyridoneinhibitor of JAK2 with high selectivity and nM efficacy [136].

A recent series of three works concerning the design and development of JAK3 selective inhibitors resulted from the joined efforts of two Chinese teams. The articles explore the design and synthesis of 1H-pyrazolo[3,4-d]pyrimidin-4-amino derivatives [138], pyrimidine-4,6-diamine derivatives [139] and 4- or 6-phenyl- pyrimidine derivatives [140], respectively. The three studies share a common pattern consisting in the design of derivatives, their in vitro evaluation and a docking analysis to better understand the interaction framework and explain the selectivity. Overall, among the examined compounds, two eventually demonstrated great affinity for JAK3 and showed high selectivity with respect to the other isoforms [139,140]. Interestingly, the docking studies pointed at Cys909 as a key residue for JAK3 selectivity, confirming the previous hypothesis according to which the aryl group of the designed derivative forms a covalent bond with the Cys909 sulfur group [139,140].

Further interesting studies concerning JAK selectivity were carried by Bajusz et al. starting in 2016 and are still on-going [141,142]. The research group succeeded in developing JAK2 selective compound with nM affinity. The initial efforts of the 2016 study consisted in the development and validation of a structure-based in silico protocol able to discriminate between an active and a decoy dataset. The method was further implemented to recognise possible hits with JAK2 over JAK1 selectivity. The main result was the discovery of an indazole derivative with a 14-fold JAK2/JAK1 selectivity and 2 mM affinity towards JAK2 [141]. 

The same compound was hence considered for optimisation in the next work. A series of derivatives was designed and synthesized and, eventually, 3-Amino-N-(2,6-dichlorophenyl)-1H-indazole-5-carboxamide was isolated as the most promising hit showing nM affinity towards JAK2 and 16-fold JAK2/JAK1 selectivity.  Noteworthy, these computation studies also revealed that the water molecules network within the binding site plays a pivotal role in the formation of ligand-protein interactions and must be hence taken into account when performing CADD [142].”

Conclusions?  There really isn’t any conclusion section.  This is where the current and the future should come together with in-depth speculation as to the current utilization of JAK inhibitors and then a deep dive into where JAK inhibitor use may be in the future.

A conclusions paragraph was added summarising the importance of the recent innovations in the field of JAK inhibitors synthesis for the most part, but also highlighting how the most up-to-date technologies can bring (and have brought)  benefits to the research of JAKis.

“Conclusions

In 2011, Ruxolitinib was approved as the first JAK inhibitor for clinical use. That first result seems so distant now, considering all the progress achieved in this field. Therefore, considerable efforts have been made to discover new molecular entities able to inhibit JAKs. Up to date, also Tofacitinib, Baricitinib, Fedratinib, and Upadacitinib have obtained FDA approval as JAKis, and several other compounds have reached the final stage of clinical trials. Lately, the involvement of JAKs in several complex diseases, including cancer, inflammation, and autoimmune diseases, has been determined. In this context, the search of new therapeutic agents as JAKis became a hot topic. These findings offered new perspectives to fight autoimmune-related diseases and severe inflammatory disorders. 

This review was organised to give an outline of the last four years’ progress in the development of JAKis. The multi-target spectrum of the “older generation” of approved JAKis, Ruxolitinib and Tofacitinib, was also discussed, given the prospects opened by this kind of approach. Nevertheless, the main focus of this review covered the aspects related to the synthesis of Ruxolitinib and Tofacitinib, thus offering an analysis from the original to the most innovative and interesting approaches for the production of these compounds.

Ruxolitinib and Tofacitinib remained the only JAKis approved in therapy for at least six years. Furthermore, most of the recent JAKis share the deazapurine core present in the Ruxolitinib and Tofacitinib scaffold. Therefore, the synthesis of the “older generation of JAKis” must not be neglected. In particular, concerning the synthesis of Ruxolitinib, only two articles and a more recent patent have been published. The patent offers a solution to some of the synthetic issues showed by the older synthetic methodologies and represents a cost-effectiveness and sustainable strategy for the industrial production of Ruxolitinib. On the other hand, the synthesis of Tofacitinib has been explored more and several synthetic procedures have been proposed. The strategies mainly differ in the preparation of the piperidine moiety, a challenging step due to the presence of two chiral centers. The first methodologies reported by Pfizer, Inc were improved during the years providing economical, efficient, and sustainable procedures more fitting with the industrial production.

Innovative technologies are the driving force of drug discovery. The recent progress in the field of computational chemistry and bioinformatics have facilitated the research protocols, providing quicker and cheaper approaches to the preliminary evaluation of new active compounds.

Computational techniques represent powerful and useful tools to reduce the time and to enhance rational drug design in the medicinal chemistry field. The combination of in silico approaches with the advances in purification and structure investigation techniques have resulted in reduced time and costs of drug development. Recent works offered a clear prospect of the benefits of SBDD for the identification of new JAK inhibitors, providing remarkable results.

The recent advances in the design and synthesis of JAKis will certainly boost the development process of new treatments offering alternative perspectives against complex diseases. “

Minor points

16           change ‘Over the last decades,’ to ‘Historically,’

17           change the wording on ‘the possibilities of JAK inhibition have been thoroughly explored’ – this is a little awkward and hard to really determine – WHAT possibilities are being explored

19-20     change the sentence ‘Considering the complexity of JAK-depending processes and their involvement in multiple disorders, JAK inhibitors are the perfect candidates for drug repurposing and for the assessment of multi-target strategies.’ To ‘JAK inhibitors are the perfect candidates for drug repurposing considering the complexity of JAK-dependent processes and their involvement in multiple disorders.’

23           change ‘the simplification’ to ‘simplify’

57           change ‘During the past years,’ to ‘Recently,’ Phrases like ‘during the past years’ or ‘nowadays’ don’t always fit within the context of the writing style

58-59    the sentence fragment ‘and the role of JAKs has gradually shifted into the spotlight, hence attracting increasing interest from the pharmaceutical industry [19-21].’ Is wordy and because of that, can be hard to follow.  Consider ‘and the role of JAKs has increased in prominence leading to increased interest from the pharmaceutical industry [19-21].’

66-69    this section should be re-written.  Adding words/phrases like ‘tip of the iceberg’, ‘kicked off the exploration’, etc. could be streamlined and simplified.  For example, instead of:

‘The increasing number of clinical trials focused on JAKis was just the tip of the iceberg; at some point, the attention of the pharmaceutical companies naturally turned towards the necessity for more efficient production strategies while researchers everywhere kicked off the exploration of this field from broader perspectives.’

Can be shortened to:

‘The increasing number of clinical trials focusing on JAKis has attracted the attention of the pharmaceutical companies, which initiated the need for more research in this area, and more efficient production strategies.’

79           change ‘proved’ to ‘suggested’

88           change ‘resulted’ to ‘was’

91           change ‘for the treatment of’ to ‘to treat’

113         change ‘Several preclinical data are available on…’ to ‘Preclinical data is available on…’

132         delete ‘the fields of’

135         change ‘fields have’ to ‘field was’

137        delete ‘nowadays’ and change ‘at the identification of’ to ‘to identify’

140         indent the first line and then readjust lines 142-144 which appear to be a hanging indent.  Change ‘When compared with…’ to ‘Compared with…’

147         delete ‘in particular’

151         change ‘which’ to ‘that’ and change ‘active’ to ‘in activity’

161         delete ‘various components’ and change ‘system’ to ‘systems’

165         fix indent and then change ‘the treatment of’ to ‘treating’

180         delete the second ‘which’

183         delete ‘in a recent work,’

195-97   the sentence ‘By binding to the extracellular domain of the receptor, cytokines drive the activation by phosphorylation of JAK proteins that serve as the docking site for STATs.’ Should be rewritten.  Maybe to ‘Cytokines drive JAK activation by binding to the extracellular domain of the receptor resulting in phosphorylation of JAK proteins that will serve as the docking site for STATs.’

216         change ‘both of which include’ to ‘including’

236         the end of the sentence ‘STATs and consequently the gene transcription.’ Should be separated into 2 sentences, the ‘consequently gene transcription’ could be expanded

237         change ‘The result of these events is a significant…’ to ‘Overall, a significant…’

238         insert ‘is observed’ after ‘response’

259         delete ‘nowadays’

267         delete the second ‘to’

278-80   the sentence ‘Despite being the first JAK1 and JAK2 inhibitor introduced in therapy, as far as our knowledge goes, there are only two articles [114,115], and a more recent patent [116] which resume the state of art and provides an alternative procedure to obtain Ruxolitinib in industrial scale.’ Is very long and can be separated into at least 2 sentences.  For example: ‘Despite being the first therapeutic JAK1 and JAK2 inhibitor, only two articles have been published [114,115]. Recently, a patent [116] which provides an alternative procedure to obtain Ruxolitinib on an industrial scale.’

293         delete ‘in turn’

308         what does ‘application to provide Ruxolitinib’ mean?  Seems like there is something missing there.

330-31   change ‘for the synthesis of’ to ‘to synthesize’

333         delete ‘most of the time’ and change ‘subjected to’ to ‘usually’

337         change ‘which provided’ to ‘providing’

338         change ‘a new method for the preparation of Ruxolitinib which was suitable for industrial’ to ‘a new method for the preparation of Ruxolitinib, which was suitable for industrial’

349         I believe ‘oxalylchloride’ is misspelled.

390        delete ‘and in order’

428         change ‘the synthesis of’ to ‘synthesizing’

433         change ‘process’ to ‘processes’

444         change ‘commercial’ to ‘commercially’

449         insert ‘a’ before ‘catalyst’

455         the sentence ‘…resolving agents, increasing the costs and time-consuming among other.’ Can be reworded for clarity to ‘…resolving agents, increasing the cost and time of synthesis.’

461         change ‘in’ to ‘of’

468         insert ‘the’ before ‘presence’

490         change ‘were’ to ‘was’

492-93   the sentence ‘According to the literature data, compared with the first syntheses published for both compounds, several synthetic constraints have been faced.’ Is unclear – this should be rewritten and maybe expanded a little for the improvement in clarity.

495         delete ‘the’

All suggestions have been changed and accepted.

Reviewer 2 Report

This is a well organized and written review about JAK Inhibitors. I feel that scientific community will be benefit from this work. I included minor english grammar edits.

Do the JAK inhibition is benefit from structure-based drug design? If yes can the authors provide an example from the literature?

What it seems to be that the authors should change is the following: since the first 10 out of 18 pages of the review is related with biochemistry and structure of JAK protein targets the conclusions of the authors should be related not only to the improvement of synthetic strategies but also to the progress related to the JAK inhibition biochemistry and the JAKs structural biology.

Author Response

Answers to Reviewer 2

This is a well organized and written review about JAK Inhibitors. I feel that scientific community will be benefit from this work. I included minor english grammar edits.

We thank the reviewer for the appreciation and we are glad that our work resulted interesting.

Do the JAK inhibition is benefit from structure-based drug design? If yes can the authors provide an example from the literature?

The authors added a whole new section 6. about structure-based drug design studies concerning JAK inhibition.

What it seems to be that the authors should change is the following: since the first 10 out of 18 pages of the review is related with biochemistry and structure of JAK protein targets the conclusions of the authors should be related not only to the improvement of synthetic strategies but also to the progress related to the JAK inhibition biochemistry and the JAKs structural biology.

The conclusions summarize the key points of each main section of the work highlighting the main results obtained in the recent years in the fields of synthesis and drug design.

Minor points

All suggestions in the peer reviewed pdf were considered and accepted.

Round 2

Reviewer 1 Report

I see no additional changes that need to be made.  I have to admit - I love the new title.  Accurate to the presentation and an attention grabber.  I appreciate the consideration and the work the authors put in on the revision.  All the best

Reviewer 2 Report

I am happy that the authors added a sub-section which describes computational drug design efforts in the field. I am happy with the review and recommend publication with no additional changes.